# On the Knowledge and Prescription of Probiotics by Pediatric Providers: A Cross-Sectional Study

**DOI:** 10.3390/nu17060963

**Published:** 2025-03-10

**Authors:** Uzma Rani, Julie Ehrlich, Ghina Fakhri, Mohammed Doklaijah, Telisa Stewart, Winter Berry, Aamer Imdad

**Affiliations:** 1State University of New York Upstate Medical University, Syracuse, NY 13210, USA; uzma-rani@uiowa.edu (U.R.); julieehrlich01@gmail.com (J.E.); fakhrig@upstate.edu (G.F.); doklaijm@upstate.edu (M.D.); stewartt@upstate.edu (T.S.); berryw@upstate.edu (W.B.); 2Division of General Pediatrics and Stead Family, Department of Pediatrics, Carver College of Medicine, University of Iowa, Iowa City, IA 52242, USA; 3Department of Pediatrics, The University of Vermont Medical Center, Burlington, VT 05401, USA; 4Division of Gastroenterology, Hepatology, Pancreatology, and Nutrition, Stead Family Department of Pediatrics, University of Iowa, Iowa City, IA 52242, USA

**Keywords:** probiotics, pediatric healthcare providers, knowledge

## Abstract

Objective: The therapeutic or prophylactic efficacy and safety of probiotics are not well established. The objective of this study was to assess the knowledge and practice of probiotic use in children among pediatric providers. Methods: This was a cross-sectional study of pediatric providers. A survey was sent to the members of the American Academy of Pediatrics, New York Chapter 1. Results: We received 168 responses. Participants were mostly females (70%) and with MD or equivalent education (93%). About 50% of responders did not select the correct definition of probiotics and confused probiotics with prebiotics and synbiotics. About 97% of practitioners were asked about the merits of probiotics by families, and 60% of respondents had prescribed probiotics in their clinical practice. The most common indication for prescription was for treatment of antibiotic-associated diarrhea. When asked about their recommendation for a family who had already started probiotics, 66% of the providers recommended continuing the probiotics. There was a significant association between the frequency of probiotics prescription and the type of practice (*p* < 0.05). However, this association disappeared after adjusting for age, gender, education, and years of practice. The more experienced the practitioner, the lower the odds were of prescribing probiotics (*p* < 0.05). Conclusions: There was inadequate knowledge about probiotics among general pediatric providers. Of the pediatricians asked about probiotics, most recommended continuing them if a family was using probiotics for a specific condition. Studies with a larger nationally representative sample are required for future research.

## 1. Introduction

The International Scientific Association for Probiotics and Prebiotics (ISAPP) defines probiotics as “live microorganisms that, when administered in adequate amounts, confer a health benefit on the host” [1]. Probiotics are used as therapeutic or preventive treatments for a variety of medical conditions including acute gastroenteritis, infantile colic, antibiotic-associated diarrhea, irritable bowel syndrome, ulcerative colitis, and prevention of neonatal sepsis and necrotizing enterocolitis in premature infants, atopic dermatitis, and asthma [2,3,4]. Probiotics are widely available commercially, but, because they are classified as food supplements, they are not regulated by the US Food and Drug Association (FDA) as a drug. Several studies have been published to evaluate the effect of probiotics on the conditions mentioned above, but the data are inconsistent [5,6,7,8]. Several organizations have issued guidelines about indications for probiotics that showed that there are very few circumstances where the use of probiotics is supported by convincing evidence of benefits [2,4,9,10]. Even though the efficacy and safety of probiotics are not well established, the use of probiotics by the public has increased; it has become a multibillion-dollar industry worldwide [11]. The few studies that have been performed evaluating the knowledge and perceptions of healthcare providers have found that healthcare providers have limited knowledge of probiotics, especially providers in developing countries [12,13]. Additionally, knowledge and perceptions vary greatly depending on the discipline of medical practice [14]. There are limited data from the US about the knowledge, attitude, and practice of probiotics among pediatric healthcare providers. This study aims to assess the knowledge and practice of probiotics use in children among pediatric providers.

## 2. Research Methodology

### 2.1. Study Design and Settings

This is a cross-sectional study to survey pediatric healthcare providers about their knowledge and practice of probiotics A study protocol was developed and registered online in the Research Registry (ID No. 8950) before the study commenced [15]. Our primary outcome was the prescription of probiotics by pediatric healthcare providers. This study’s findings are reported according to STROBE guidelines for observational studies [16].

### 2.2. Study Population

The study population was pediatric healthcare providers, including pediatric residents, physicians, pediatric nurse practitioners, and physician assistants. Participants were contacted by sending emails to members of the American Academy of Pediatrics (AAP), New York State Chapter 1, and practitioners listed in the Pediatrics Department email list at SUNY Upstate Medical University.

### 2.3. Research Instrument

A questionnaire was developed based on the survey used by Fijan et al. in 2019 [14]. The survey was divided into three broad sections by topic. The first section comprised five questions related to *demographic characteristics*. These included age, gender, education, number of years of practice, and type of practice. The next three questions focused on *knowledge* and the source of knowledge of probiotics. The last section of the survey was focused on the *perception* and *practice* of probiotic use among pediatric healthcare providers. Eight questions were included in this section. Participants were encouraged to provide free-text comments on probiotics in the last question. The questionnaire was pilot-tested on 10 pediatric providers for their feedback about the readability and understandability of the questionnaire. Their feedback was incorporated into the refinement of the questionnaire. The Redcap survey tool was used to distribute the survey questions and collect the data.

### 2.4. Ethical Consideration

This study was approved by the SUNY Upstate Medical University institutional review board (IRB) for exemption (IRB ID No. 2041214). No identifying information was collected from the participants. The survey was sent to the AAP Chapter 1 email list to mitigate selection bias.

### 2.5. Data Collection and Storage

The survey was distributed via emails with a Redcap survey link to American Academy of Pediatrics members, New York Chapter 1, and practitioners on the SUNY Upstate Department of Pediatrics email list. The survey link stayed open from 22 May 2023 to 30 June 2023. Two email reminders were sent to non-responders to complete the survey. All data were stored in the institutional Redcap online platform. The data files were downloaded to a password-protected computer for statistical analysis. The data file is available on request from the corresponding author.

### 2.6. Method of Data Analysis

Data were cleaned and coded for further analyses using SPSS V 28.0 software [17]. Descriptive analysis was performed for all included variables. Categorical variables were reported as count and percentage. No continuous variables were collected in the data. Bivariate analysis was performed by using a Chi-square test for association between the outcome variable, “probiotic prescription”, and exposure demographic variables (age, gender, education, years of practice, and type of practice). Logistic regression analysis was performed to assess the association between independent demographic variables and the dichotomous dependent outcome variable of probiotic prescription, and results are reported as odd ratios (ORs), along with 95% confidence intervals (CIs). Variables included in the logistic regression model were selected based on theoretical relevance and previous research indicating their potential association with probiotic prescription practices. Variables such as age, gender, education, years of practice, and type of practice were included to control for potential confounding factors. The final logistic regression model was selected using a stepwise approach, with variables retained based on their statistical significance (*p* < 0.05) and contribution to the model’s overall fit. Statistical significance was set at a *p*-value less than 0.05. Subjects with missing responses for the variables were excluded from the final logistic regression model and the final number of participants was noted to be 151.

## 3. Results

### 3.1. Study Population Demographics

The total sample size was 168. The participants were mostly female (70%), less than 60 years of age (74%), with an MD or equivalent education (93%) (Table 1). One-fifth of the participants had less than 5 years of practice experience. About 21% had 11–20 years, 23% had 21–30 years, and another 23% had >30 years of practice experience. A total of 46% percent of participants worked in private practice, 42% worked in academic practice only, and the rest worked in a combination of academic and private practice.

### 3.2. Knowledge About Probiotics

About half of the participants did not identify the correct definition of “probiotics”; however, most (73%) were aware that probiotics are dietary supplements that are not regulated as drugs by the FDA. The most common source of information about probiotics was medical journals (33%), followed by clinical training (22%) and Google and other search engines (15%) (Table 2).

### 3.3. Practice of Probiotics

Ninety-seven percent of providers stated that families have asked them about probiotics. About 62% of the pediatric providers had prescribed probiotics with variable frequency. The most common indication was treatment of antibiotic-associated diarrhea, followed by treatment of acute diarrhea and prevention of *Clostridioides difficile* infection (Figure 1). About 57% of the prescribers had no preference for single or multiple strains of probiotics. When asked if they would recommend a formula brand that has probiotics inside, 17% of the participants confirmed that they would. Over-the-counter probiotics were recommended by 82% of the participating pediatric providers. When asked about their recommendation to a family who had already started probiotics, 68% of the pediatric providers recommended continuation of the probiotics (Table 2).

Bivariate analyses (Table 3) showed no significant association between probiotic prescription and the provider’s age (*p* = 0.14), gender (*p* = 0.27), education (*p* = 0.15), and years of practice (*p* = 0.08). There was a statistically significant association between the prescription of probiotics and the type of practice (*p* < 0.05). However, when controlled for age, gender, education, and years of practice, there was no statistically significant difference between providers working in academic institutions or combined practices when compared to private practice, (OR = 1.2, 95% CI: 0.59 to 6.06), and (OR = 0.76; 95% CI 0.26 to 11.79), respectively. Values with the missing data were excluded in the logistic regression model (final N = 151).

Increasing years of experience was associated with lower odds of prescribing probiotics. After controlling for age, gender, education, and practice type, the providers with between 20 and 30 years of practice and those with more than 30 years of practice had lower odds of prescribing probiotics when compared to providers with less than 5 years of experience, (OR = 0.12; 95% CI 0.02 to 0.91) and (OR = 0.15; 95% CI 0.03 to 0.74), respectively. After excluding the subjects with missing responses of included variables, a sample size of 151 was included in the logistic regression. Additional analyses based on categories of age, gender (male vs. female), and education (MD/equivalent vs. nurse practitioner and physician assistant) did not show a statistically significant difference related to the prescription of probiotics (Table 4).

## 4. Discussion

This cross-sectional study on knowledge and practice of probiotics showed that about half of the participants did not know the correct definition of probiotics. Most of the pediatric providers had been asked about probiotics in their practice, which may point to an increased curiosity about probiotics among parents and families. Our study results also indicate that pediatric providers in private practice are more likely to prescribe probiotics, although these results were not statistically significant on logistic regression after accounting for potential confounding factors. The most common indication for the prescription of probiotics was for antibiotic-associated diarrhea. Most providers recommended over-the-counter probiotics.

There are limited data on the knowledge and prescription patterns of probiotics in the United States, particularly among pediatric healthcare providers. Our study results are in line with previous studies conducted outside of the USA, reporting the widespread use of probiotics among the general population and limited knowledge of healthcare providers. Fijan et al. performed an online study with participants from 30 countries to assess the knowledge of probiotics among healthcare providers [14]. The study results showed that while only 8.9% of providers rated their knowledge as excellent, 80% of the participants were able to identify the correct definition of probiotics [14]. Our study showed that about half the participants were not able to identify the correct definition of probiotics and confused them with prebiotics and synbiotics. This finding points to inadequate knowledge and training among pediatric care providers. This is yet reiterated by a finding noted in our study as well. Our participants have reported that if probiotics were started by another provider for their patients, there is discomfort regarding stopping them or recommending discontinuing them. This stems from significant discomfort regarding this topic, and lack of knowledge and evidence-based medicine regarding its proper uses, duration of therapy when indicated, and side effects. Another study by Pettoello-Mantovani et al. from Europe showed that half of the pediatricians and more than 90% of the dietitians rated their knowledge and training about probiotics as “some/a little” [18]. On the other hand, there was a great desire for probiotics information by the public in the same study. Ninety-seven percent of the participants in our study stated that families had asked them about probiotics.

The possible explanations for the limited knowledge of pediatric care providers are inadequate available resources and conflicting evidence related to efficacy, indications, and adverse effects of probiotics. For example, the most common indication for probiotics is for the prevention and treatment of antibiotic-associated diarrhea and acute gastroenteritis. Studies from low- and middle-income countries, especially those that use probiotic preparations containing *Saccharomyces boulardii* [17] and *Lactobacillus*, showed that probiotics shorten the duration of diarrhea in children by one day [19,20]; however, randomized controlled trials from high-income countries did not show a clinical benefit of probiotics on the symptoms or duration of diarrhea [6,21,22]. Concerns have also been raised about the safety of probiotics when used for the prevention and treatment of antibiotic-associated diarrhea. In their study, Seuz et al. pointed to persistent, long-term, probiotics-induced dysbiosis after a comparison of probiotic treatment and no-intervention groups for antibiotic-induced diarrhea [5]. Similarly, conflicting evidence exists for the beneficial effects of probiotics in *Clostridioides difficile*-associated diarrhea, irritable bowel syndrome, necrotizing enterocolitis, and acute respiratory infections [4,9,10,22]. The mixed evidence underscores the need for extensive, high-quality research on probiotics, focusing on both in vitro and in vivo studies. This will help to create evidence-based recommendations and clinical practice guidelines for pediatric care providers and help them to give consistent guidance to patients and their families.

Our study highlights the fact that well-designed interventions are needed for healthcare provider education with a focus on increasing provider knowledge about probiotics and their use. This can prepare healthcare providers to advise the public about over-the-counter probiotics and the evidence-based indications that guide the prescription of probiotics.

### Limitation of the Study

This was a cross-sectional study that involved pediatric healthcare providers from the New York State area, and, therefore, the results from our study cannot be generalized. In addition, we have excluded the data points with missing information, which increases the risk of information bias. While there have been data on probiotics, indications as well as benefits, there remains a significant need for studies with larger sample sizes dedicated to pediatrics. The foundation for these studies should start with a documented lack of knowledge from providers to allow for research funding on this important topic.

## 5. Conclusions

This study concluded that pediatricians had limited knowledge regarding probiotics, their uses, indications, and side effects. However, pediatricians were commonly asked about probiotics, and most recommended continuation if a family was using probiotics for a certain condition. Larger studies with a representative national sample will be required for future research to identify the trends in the use of probiotics in the pediatric population.

## Figures and Tables

**Figure 1 nutrients-17-00963-f001:**
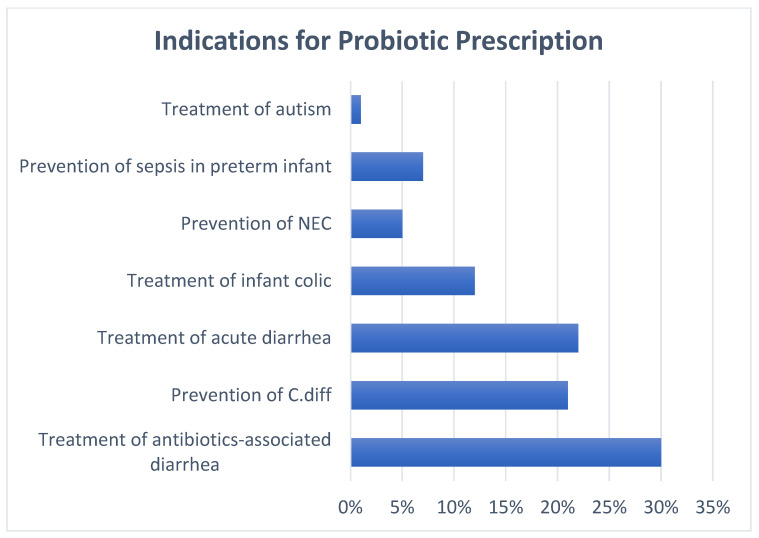
Indications for probiotic prescription. Participants were allowed to select multiple indications for this question. Abbreviations: NEC; necrotizing enterocolitis, C. Diff; *Clostridioides difficile*.

**Table 1 nutrients-17-00963-t001:** Demographic characteristics of respondents.

	Count	Percent Overall (%)	Valid Percent (%)
Total Sample	168	100	100
Age (Years)		100	100
20–29	14	8.3	8.4
30–39	39	23.2	23.5
40–49	39	23.2	23.5
50–59	31	18.5	18.7
>60	43	25.6	25.9
Missing	2	1.2	
Sex	168	100	100
Male	50	29.8	30.3
Female	115	68.5	69.7
Missing	3		
Education	168	100	100
MD or equivalent	155	92.3	92.8
Other *	12	7.1	7.2
Missing	1	0.6	
Years of Practice	168	100	100
Less than 5 years	38	22.6	23.4
5–10	17	10.1	10.4
11–20	34	20.2	20.7
21–30	37	22	22.6
>30	38	22.6	23.2
Missing	4	2.4	
Practice Type	168	100	100
Private practice only	76	45.2	46.3
Academic practice only	68	40.5	41.5
Both academic and private	20	11.9	12.2
Missing	4	2.4	

* Other: includes nurse practitioners, physician assistants, and medical students.

**Table 2 nutrients-17-00963-t002:** Summary of knowledge and prescription practice of probiotics among the respondents.

**Knowledge**	**% (Count)**
Definition of probiotics	
Nondigestible food components	2.6 (4)
Live microorganisms	50.6 (79)
Both ‘a’ and ‘b’	38.5 (60)
Neither	3.2 (5)
I am unsure of the answer	5.1 (8)
Regulation of probiotics	
FDA approved	1.3 (2)
Dietary supplements	73.2 (115)
Neither is true	25.5 (40)
Source of information about probiotics	
Google and other search engines	14.6 (23)
Clinical training	21.7 (34)
Medical journals	33.1 (52)
Medical conferences	10.2 (16)
Specialist consultation	6.4 (10)
I have not obtained info	14 (22)
**Practice**	
How frequently do families ask you about probiotics?	
Very frequently	2.6 (4)
Frequently	19.2 (29)
Occasionally	45.7 (69)
Rarely	21.2 (32)
Very rarely	7.9 (12)
Never	3.3 (5)
Have you ever prescribed probiotics?	
No	38.4 (58)
Yes	61.6 (93)
How often do you prescribe probiotics?	
1–2 times per year	17.2 (16)
1–2 times per month	33.3 (31)
Many times, a week	15.1 (14)
Once a week	15.1(14)
Rarely	19.4 (18)
Do you prefer to prescribe probiotics that have	
Single strain	14 (13)
Multiple strain	29 (27)
No preference	57 (53)
Have you recommended infant formula with probiotics for any indication?	
No	82.9 (121)
Yes	17.1 (25)
Have you ever recommended OTC probiotics?	
No	18.5 (27)
Yes	81.5 (119)
If a family informs you that the family has been giving their child probiotics, what is your typical response?	
Probiotics are beneficial + no side effects-→continue	53.7 (80)
Probiotics are not beneficial + no side effects-→continue	14.1 (21)
Probiotics are not beneficial + have side effects→discontinue the probiotics.	4 (6)
I don’t give any recommendations about it	28.2 (42)

**Table 3 nutrients-17-00963-t003:** Bivariate analyses of predictor variables and probiotics prescription.

	Probiotics Prescription				
Variables	No	Yes	Row Count *	X^2^ (DF) **	*p*-Value ***
Age (years)	Count (%)	Count (%)		6.945 (4)	0.14
20–29	2 (18.2)	9 (81.8)	11		
30–39	17 (48.6)	18 (51.4)	35		
40–49	17 (47.2)	19 (52.8)	36		
50–59	11(39.3)	17 (60.7)	28		
>60	11 (26.8)	30 (73.2)	41		
Gender				1.231 (1)	0.27
Male	14 (31.8)	30 (68.2)	44		
Female	44 (41.5)	62 (58.5)	106		
Education				2.110 (1)	0.15
MD and equivalent	52 (36.9)	89 (63.1)	141		
Other	6 (60)	4 (40)	10		
Years of Practice				8.393 (4)	0.08
Less than 5	14 (42.4)	19 (57.6)	33		
5–10	7 (41.2)	10 (58.8)	17		
11–20	15 (50)	15 (50)	30		
21–30	15 (44.1)	19 (55.9)	34		
>30	7 (18.9)	30 (81.1)	37		
Practice Type				6.583 (2)	<0.05
Private office only	20 (27.8)	52 (72.2)	72		
Academic institution only	29 (48.3)	31 (51.7)	60		
Both private and academic	9 (47.4)	10 (52.6)	19		

* Total row count. ** Pearson Chi-square test was performed for all categorical predictor variables, DF = degree of freedom, Y, y = years. *** *p*-value was set at equal to or less than 0.05.

**Table 4 nutrients-17-00963-t004:** Logistic regression examining predictors of probiotics prescription.

	OR	*p*-Value	95% CI
Age (years)			
20–29	reference	reference	reference
30–39	12.85	0.08	0.76, 217.40
40–49	2.42	0.46	0.23, 25.31
50–59	2.2	0.41	0.34, 14.48
>60	2.2	0.33	0.45, 10.65
Gender			
Male	reference	reference	reference
Female	1.25	0.6	0.54, 2.89
Education			
Other	reference	reference	reference
MD or equivalent	2.67	0.2	0.60, 11.79
Years of Practice			
Less than 5	reference	reference	reference
5–10 y	0.13	0.12	0.10, 1.65
11–20 y	0.24	0.27	0.02, 3.05
21–30 y	0.12	0.04	0.02, 0.91
>30 y	0.15	0.02	0.03, 0.743
Type of Practice			
Private office only	reference	reference	reference
Academic only	1.2	0.28	0.59, 6.06
Both private and academic	0.76	0.64	0.26, 11.79

OR = odds ratio, CI = confidence.

## Data Availability

Data is available upon request. Please contact the corresponding author.

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
