# Peer review of "On the Knowledge and Prescription of Probiotics by Pediatric Providers: A Cross-Sectional Study"

_nutrients, 2025, doi:10.3390/nu17060963_

Round 1
Reviewer 1 Report
Comments and Suggestions for Authors
Dear Authors,
The manuscript "On the Knowledge and Prescription of Probiotics by Pediatric Providers: A Cross-Sectional Study" presents significant scientific relevance. It shows the knowledge and practice of 19 probiotic use in children among pediatric providers. However, I have only a few comments.
- I suggest including the graphical abstract, this could increase the visibility of the article.
- Can the authors explain inconsistencies in Table 1? 166 Age Answers were used, but 165 volunteers responded to sexuality (male and female) and 167 people indicated the level of education.
- Can the authors explain inconsistencies in Table 2? About the definition of probiotics, 156 answers were exposed, but 157 volunteers answered to the regulation of probiotics.
- Figure 1 must present the data with greater clarity, indicating the percentage corresponding to autism treatment, for example. I recommend using another software to elaborate the graphic (result).
- I recommend that authors improve the number of volunteers in the manuscript, including members of other societies.
Author Response
Comment 1: I suggest including the graphical abstract, this could increase the visibility of the article.
Response 1: Thank you for your suggestion. We think that this study could be published without a graphical abstract.
Comment 2: Can the authors explain inconsistencies in Table 1? 166 Age Answers were used, but 165 volunteers responded to sexuality (male and female) and 167 people indicated the level of education.
Response 2: We have revised Table 1 to address the inconsistencies. The discrepancies in the number of responses for age, gender, and education were due to missing data. Table one now includes information about the missing responses.
Comment 3: Can the authors explain inconsistencies in Table 2? About the definition of probiotics, 156 answers were exposed, but 157 volunteers answered to the regulation of probiotics.
Response 3: The inconsistencies in Table 2 regarding the definition of probiotics and the regulation of probiotics have been clarified. The discrepancy was due to one responder not answering the question, which has been accounted for in the revised table.
Comment 4: Figure 1 must present the data with greater clarity, indicating the percentage corresponding to autism treatment, for example. I recommend using another software to elaborate the graphic (result).
Response 4: We have improved the quality of Figure 1 to present the data with greater clarity. We have recreated the image.
Comment 5: I recommend that authors improve the number of volunteers in the manuscript, including members of other societies.
Response 5: We agree that increasing the number of volunteers and including members of other societies would be beneficial. This is a valuable suggestion for future research projects.
Reviewer 2 Report
Comments and Suggestions for Authors
The manuscript titled "On the Knowledge and Prescription of Probiotics by Pediatric Providers: A Cross-Sectional Study" presents an interesting and relevant investigation into pediatric providers' knowledge and practices regarding probiotics. While the study addresses an important topic, several critical issues need to be addressed before the manuscript is suitable for publication.
Major Concerns:
1.High Frequency of Typographical and Grammatical Errors:
The manuscript contains numerous typographical errors, awkward sentence constructions, and grammatical mistakes, which significantly affect readability. I have identified and listed all mistypes below, and the authors must thoroughly proofread and revise the text for clarity and correctness.
2.Low-Quality Figures:
The authors must ensure that all figures are of high resolution, properly labeled, and formatted for readability. If applicable, raw data or better-formatted charts should be provided.
3.Issues with Writing Clarity and Flow:
Several sentences are structured awkwardly, making it difficult to understand the intended meaning. The authors should revise the manuscript for better readability and coherence.
4.Methodological and Data Analysis Concerns:
he study mentions that it followed STROBE guidelines, but some methodological details (e.g., how response bias was addressed, how non-respondents were accounted for) require clarification. The logistic regression model lacks an explanation of why certain variables were included or excluded.
List of Mistyped Words and Errors Identified:
Below are some of the typographical errors found in the manuscript:
1.Abstract and Introduction:
"therapeutic or prophylactic efficacy and safety of probiotics are not well established.The objective..."
→ Space needed after the period: "...not well established. The objective..."
"Half of the respondents did not select the correct definition of probiotics and confused them with prebiotics and synbiotics."
→ Consider rephrasing for clarity.
"Studies with a representative national sample will be required for future research."
→ "Studies with a nationally representative sample are required for future research."
2.Methods:
"A study protocol was developed before the start of the study and was registered online on the Research Registry (ID No. 8950)."
→ "A study protocol was developed and registered online in the Research Registry (ID No. 8950) before the study commenced."
"Survey was sent to AAP Chapter 1 email list in order to address the selection bias."
→ "The survey was sent to the AAP Chapter 1 email list to mitigate selection bias."
3.Results:
"About 21% had 11-20 years, 23% between 21-30 years and antoher 23% with >30 years of practice experience."
→ "About 21% had 11-20 years, 23% had 21-30 years, and another 23% had >30 years of practice experience."
→ "antother" → "another"
"We also encouraged the participants to add comments about probiotics in free-text form in our last question."
→ "Participants were encouraged to provide free-text comments on probiotics in the last question."
4.Discussion:
"There is limited data on knowledge and prescription patterns of probiotics in the United States, particulary in pediatric healthcare providers."
→ "There is limited data on the knowledge and prescription patterns of probiotics in the United States, particularly among pediatric healthcare providers."
→ "particulary" → "particularly"
"The mixed evidence underscores the need of extensive, high-quality research on probiotics with a focus on both in vitro and in vivo studies..."
→ "The mixed evidence underscores the need for extensive, high-quality research on probiotics, focusing on both in vitro and in vivo studies."
5.Conclusions:
"This study concluded that pediatricians had limited knowledge regarding the probiotics, their uses and indications as well as their side effects."
→ "This study concluded that pediatricians had limited knowledge regarding probiotics, their uses, indications, and side effects."
Author Response
Major Concerns:
Comment 1.High Frequency of Typographical and Grammatical Errors:
The manuscript contains numerous typographical errors, awkward sentence constructions, and grammatical mistakes, which significantly affect readability. I have identified and listed all mistypes below, and the authors must thoroughly proofread and revise the text for clarity and correctness.
Response 1: We have reviewed the manuscript thoroughly and corrected all typographical errors, awkward sentence constructions, and grammatical mistakes. Thank you for identifying these issues.
Comment 2.Low-Quality Figures:
The authors must ensure that all figures are of high resolution, properly labeled, and formatted for readability. If applicable, raw data or better-formatted charts should be provided.
Response 2: We have revised the figures to ensure they are of high resolution, properly labeled, and formatted for readability.
Comment 3. Issues with Writing Clarity and Flow:
Several sentences are structured awkwardly, making it difficult to understand the intended meaning. The authors should revise the manuscript for better readability and coherence.
Response 3: The manuscript has been proofread to improve readability and coherence. Awkward sentence structures have been revised for better clarity.
Comment 4. Methodological and Data Analysis Concerns:
The study mentions that it followed STROBE guidelines, but some methodological details (e.g., how response bias was addressed, how non-respondents were accounted for) require clarification. The logistic regression model lacks an explanation of why certain variables were included or excluded.
Response 4: Thank you for identifying this. Following text is added in the manuscript for clarification. "Two email reminders were sent to non-responders to complete the survey. Variables included in the logistic regression model were selected based on theoretical relevance and previous research indicating their potential association with probiotic prescription practices. Variables such as age, gender, education, years of practice, and type of practice were included to control for potential confounding factors. The final logistic regression model was selected using a stepwise approach, with variables retained based on their statistical significance (p<0.05) and contribution to the model's overall fit."
Comment 5: List of Mistyped Words and Errors Identified:
Below are some of the typographical errors found in the manuscript:
1.Abstract and Introduction:
"therapeutic or prophylactic efficacy and safety of probiotics are not well established.The objective..."
→ Space needed after the period: "...not well established. The objective..."
"Half of the respondents did not select the correct definition of probiotics and confused them with prebiotics and synbiotics."
→ Consider rephrasing for clarity.
"Studies with a representative national sample will be required for future research."
→ "Studies with a nationally representative sample are required for future research."
2.Methods:
"A study protocol was developed before the start of the study and was registered online on the Research Registry (ID No. 8950)."
→ "A study protocol was developed and registered online in the Research Registry (ID No. 8950) before the study commenced."
"Survey was sent to AAP Chapter 1 email list in order to address the selection bias."
→ "The survey was sent to the AAP Chapter 1 email list to mitigate selection bias."
3.Results:
"About 21% had 11-20 years, 23% between 21-30 years and antoher 23% with >30 years of practice experience."
→ "About 21% had 11-20 years, 23% had 21-30 years, and another 23% had >30 years of practice experience."
→ "antother" → "another"
"We also encouraged the participants to add comments about probiotics in free-text form in our last question."
→ "Participants were encouraged to provide free-text comments on probiotics in the last question."
4.Discussion:
"There is limited data on knowledge and prescription patterns of probiotics in the United States, particulary in pediatric healthcare providers."
→ "There is limited data on the knowledge and prescription patterns of probiotics in the United States, particularly among pediatric healthcare providers."
→ "particulary" → "particularly"
"The mixed evidence underscores the need of extensive, high-quality research on probiotics with a focus on both in vitro and in vivo studies..."
→ "The mixed evidence underscores the need for extensive, high-quality research on probiotics, focusing on both in vitro and in vivo studies."
5.Conclusions:
"This study concluded that pediatricians had limited knowledge regarding the probiotics, their uses and indications as well as their side effects."
→ "This study concluded that pediatricians had limited knowledge regarding probiotics, their uses, indications, and side effects."
Response: The identified typographical errors have been corrected in the manuscript. Below are some examples of the revisions made.
Round 2
Reviewer 1 Report
Comments and Suggestions for Authors
Dear Authors,
I appreciate your response.
Regards.
Author Response
Thank you.
Best regards.
Reviewer 2 Report
Comments and Suggestions for Authors
The authors have thoroughly addressed my concerns and questions.
Author Response
Thank you.